# Study on the Drag Reduction Characteristics of the Surface Morphology of *Paramisgurnus dabryanus* Loach

**Liyan Wu, Jiaqi Wang, Guihang Luo, Siqi Wang, Jianwei Qu, Xiaoguang Fan and Cuihong Liu \***

Mechanical Department, Engineering College, Shenyang Agricultural University, Shenyang 110866, China; wly78528@syau.edu.cn (L.W.); jiaqiwang202108@163.com (J.W.); guihangluo03@163.com (G.L.); wangsiqi0225@163.com (S.W.); qujianwei1986@163.com (J.Q.); xiaoguangfan1982@syau.edu.cn (X.F.)
\* Correspondence: cuihongliu77@syau.edu.cn

**Abstract:** The drag reduction design of underwater vehicles is of great significance to saving energy and enhancing speed. In this paper, the drag reduction characteristics of *Paramisgurnus dabryanus* loach was explored using 3D ultra-depth field microscopy to observe the arrangement of the scales. Then, a geometric model was established and parameterized. A simulated sample was processed by computer numerical control (CNC) machining and tested through using a flow channel bench. The pressure drop data were collected by sensors, and the drag reduction rate was consequently calculated. The test results showed that the drag reduction rate of a single sample could reach 23% at a speed of 1.683 m/s. Finally, the experimental results were verified by numerical simulation and the drag reduction mechanism was explored. The boundary layer theory and RNG *k-ε* turbulence model were adopted to analyze the velocity contour, pressure contour and shear force contour diagrams. The numerical simulation results showed that a drag reduction effect could be achieved by simulating the microstructure of scales of the *Paramisgurnus dabryanus* loach, showing that the results are consistent with the flow channel experiment and can reveal the drag reduction mechanism. The bionic surface can increase the thickness of boundary layer, reduce the Reynolds number and wall resistance. The scales disposition of *Paramisgurnus dabryanus* loach can effectively reduce the surface friction, providing a reference for future research on drag reduction of underwater vehicles such as ships and submarines.

**Keywords:** *Paramisgurnus dabryanus* loach; scales; microscopic morphology; drag reduction characteristics; bionic design



## 1. Introduction

For underwater vehicles, the frictional resistance from water accounts for about 80% of the total resistance [1]. In order to improve the power efficiency, it is necessary to reduce the surface friction of underwater vehicles. The theory of bionic tribology has become popular in the field of drag reduction in recent decades. There are two main factors affecting the tribological characteristics of a surface: the composition of the material itself and the surface structure [2–5]. Hossein [6] used two different $SiO_2$ nanoparticle surfaces, modified with polydimethylsiloxane (PDMS) and beeswax. The drag reduction of the surfaces could reach up to 24%. Ganesh [7] fabricated rice-shaped $TiO_2$ nanostructures by an electro-spinning technique to create a solid superamphiphobic coating on glass substrates. Changing the material properties or adding smooth coatings may change the original material's performance and require high costs [8]. Therefore, the research on microstructure drag reduction has become increasingly popular, and it has been applied in the fields of natural gas pipelines, airplanes, navigation, swimming suits, etc. [9–11].

Through billions of years of selection, organisms in Nature have evolved almost perfect functions and structures. They compete with each other and consequently form the most reasonable, economical and effective drag reduction function which provides researchers with natural samples and principles to invent novel drag reduction techniques [12].

The microstructure plays an important role in the drag reduction effect. Scholars have explored various preparation methods for different structures, such as direct biological replication and micro-imprint technology, photolithography, laser engraving and polishing, CNC machining, 3D printing technology, electrospinning technology, etc. [13–15]. Lang [16] printed small transverse grooves of both rectangular and sinusoidal shapes by 3D technology and found that the sinusoidal grooves which most resembled dolphin skin were proved the most effective in drag reduction. Luo [17] used silicone rubber and resin polymer to vacuum-cast shark skin and obtained a flexible biomimetic shark skin through repeated demolding, and the peak of drag reduction rate could reach 14%. Kim [18] created overhanging arrays of micro-disks through a simple lithographic technique to form a cylindrical channel, which is promising for biofouling-free tubing for blood circulation. Li [19] manufactured five types of bionic flexible coatings that mimic the skin of dolphins, and the drag reduction rate was 21.6% when the rotation speed of the rotating aluminum disc was at 50 r/min. By a 3D printing method, Graeber [20] prepared-combined arrays of alternating surface protrusions and indentations. Lloyd [21] printed two arrays of denticles on a flat plate and obtained a drag reduction of 2%. The bionic studies on massive aquatic creatures include those on rectangular, V-shaped, U-shaped and L-shaped biomimetic shark scales [22,23]. There are also some studies on waveform non-smooth surfaces [24–27]. For smaller or slower fishes, there are few studies on the distribution of surface scales and the characteristics of surface flow field.

Numerical simulation of flow fields has enjoyed more and more application in drag reduction analysis of bionic surfaces. Chen [28] established a bionic dual surface model imitating tuna skin, and found that at a speed of 6.94 m/s, the maximum drag reduction rate was 25.7%. Liu [29] obtained a maximum drag reduction rate of 9.85% for the bionic blade of L-shaped slot under braking conditions. Zhou [30] used the Fourier function to fit the non-smooth structure of the pufferfish surface and found that when the flow velocity was 5 m/s, the reduction rate of viscous resistance was 23.2%, and the total drag reduction rate was 12.94%. Mohammadi [31] proved that the triangular, trapezoidal, rectangular and circular grooves could reduce the shear resistance in a laminar flow channel driven by gradient pressure. Rong [32] designed a periodic array micro-nanostructure of bionic fish scales. The surface of fish scales is impregnated with lubricant oil and the ambient flow medium is water. At a flow velocity of 2 m/s, the drag reduction rate along and in the reverse direction of the fish scales were 33.86% and 28.95%, respectively.

Loaches can usually swim at speeds of up to 3.0 m/s and even faster when they are being hunted [33]. In this study, fan-shaped units were proposed based on the scale shape of loaches. The microstructure of loach skin was taken as the size parameter. Bionic units were processed on the aluminum substrate by CNC machining. Resistance tests were carried out on a flow channel test bench. Finally, the experimental results were verified by numerical simulation and the drag reduction mechanism was analyzed. This research can provide a reference for drag reduction of underwater vehicles.

## 2. Materials and Methods

### 2.1. Analysis of the Scale Structure of the Loach

First, the loach was anesthetized with diethyl ether (Fuyu Chemical Company Limited, Tianjin, China). A piece of skin was removed from the abdomen and immersed in 5% sodium hydroxide (Rui Jinte Chemicals Company Limited, Tianjin, China) solution for 2 min to remove the mucus. The treated loach skin was washed in a Petri dish with deionized water. Most of the liquid was absorbed with absorbent paper and the sample immediately observed with a VHX-5000 3D ultra-depth field microscope (KEYENCE, Osaka, Japan). The whole process involved no animal abuse or blood loss and complied with the Chinese law on the Protection of Animals. Ethical approval was given by the Animal Experimental Ethical Inspection, Shenyang Agricultural University.

## 2.2. Preparation of Samples

By using a CNC machining system, an aluminum substrate was carved into a sample with scale-like morphology and a size of $70 \times 64 \times 5$ mm$^3$. The preparation process is shown in Figure 1.

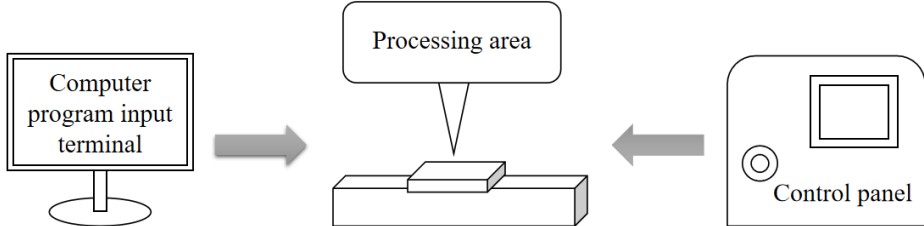

**Figure 1.** The preparation process.

Real loach scales are irregularly round, so they were simplified to a regular circle. The shape of a single loach scale exposed outside was obtained by drawing an auxiliary circle. The shape parameter (a) was defined as the center distance between the original circle and the auxiliary circle. Figure 2b,c show more structural details of the artificial sample under $200\times$ magnification. The height of the outer edge of each unit (h) is 20 μm, the diameter (d) is 0.60 mm, the shape parameter (a) is 0.36 mm, and the array distribution distance (b) is 0.43 mm.

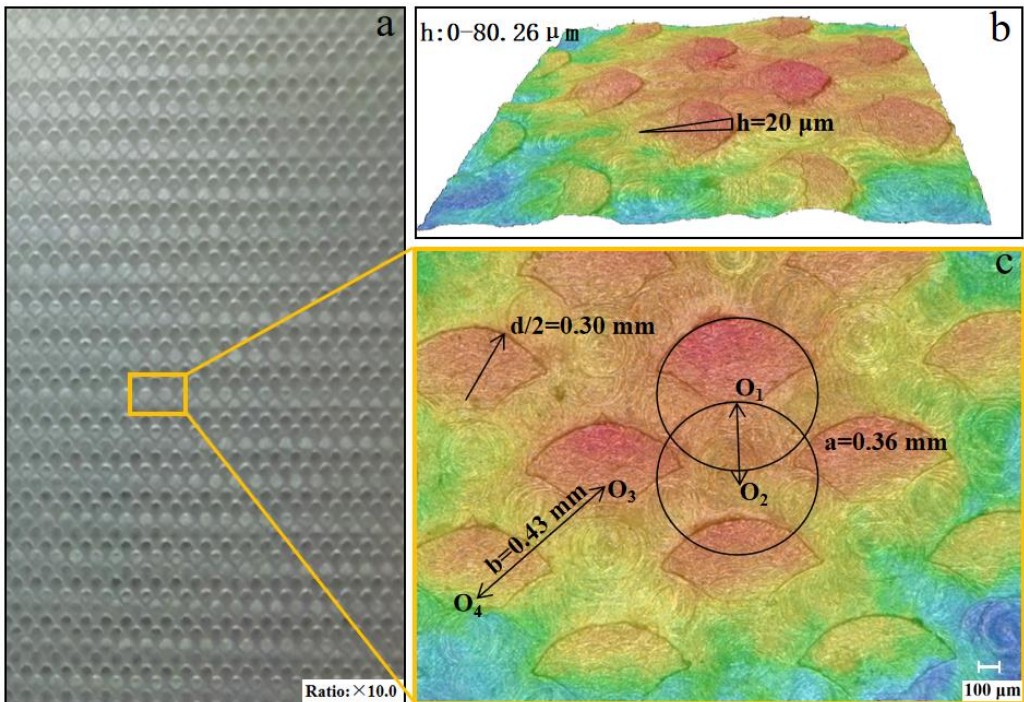

**Figure 2.** The aluminum artificial sample. (**a**) A sample with scale-like morphology; (**b**) The height "h" of the artificial sample under 200 times magnification; (**c**) The structural details of "a", "b" and "d".

## 2.3. Flow Channel Experiments

### 2.3.1. Design of Test Device

In a channel, fluid resistance is expressed by the pressure drop between two points. The resistance can be measured by collecting the pressure values between two points [34]. In this study, a flow channel was built to test the resistance. The test equipment comprises a power device (40TBFS15-10-1.5 pump, Xiepan Pump Industry, Shanghai, China), a flow regulating device (T40H-16P valve, Huizheng Automatic valve Group Company

Limited, Wenzhou, China), a flow velocity detecting device (Tianxing Shengshi Technology Company Limited, Beijing, China), a pressure detecting device (TXY815 pressure sensor, Tianxing Shengshi Technology Company Limited, Beijing, China), a data processing device (NI9203 Data Acquisition Card (DAQ card), National Instruments, Austin, TX, USA), a circulating pipeline and a tank (Figure 3). By adjusting the valve, the resistance on the surface of the samples under different flow velocity was tested.

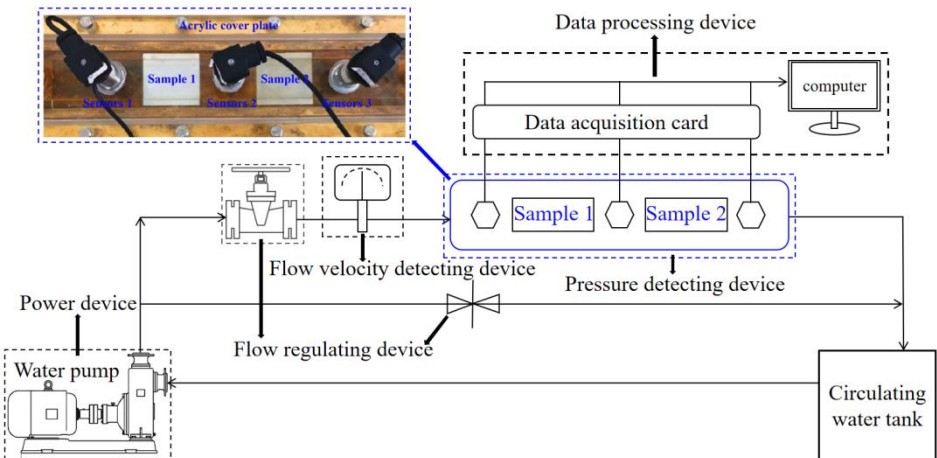

**Figure 3.** Flow channel.

Details of the facilities are listed in Table 1.

**Table 1.** Type, accuracy and range of experimental equipment.

| Equipment | Type | Accuracy | Range |
|---|---|---|---|
| Pressure sensor | TXY815 | 0.5% | 0–0.6 MPa |
| Flow meter | TXY920 | Level 0.5 | 0–18 m$^3$/h |
| DAQ card | NI9203 | 16-bit resolution 200 k/s sample frequency | - |

The external dimensions of the copper base of the pressure detecting device are 580 mm length and 150 mm width, the inner area of the flow passage is 500 mm length, 70 mm width and 8 mm depth, as shown in Figure 4a. In this experiment, a H-shaped frame with 5 cm thick with grooves was placed in the copper base, as shown in the Figure 4b. The samples were fixed in the corresponding grooves. A bulge of 10 mm width and 3 mm height is designed on both sides of the acrylic cover plate to compress the H frame, as shown in Figure 4c. Consequently the size of the cross-sectional area of the flow test section was 50 mm width and 3 mm height, as shown in Figure 4c.

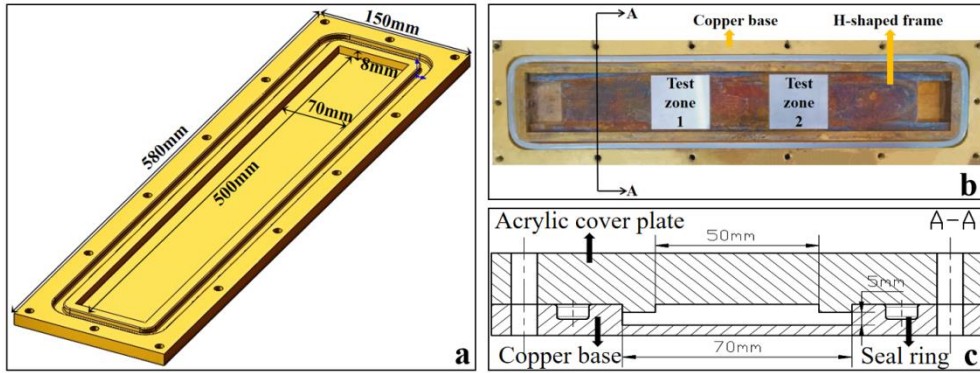

**Figure 4.** Size of the flow channel. (**a**) The size of the external of base and the inner area of the flow passage; (**b**) H-shaped frame; (**c**) The size of the cross-sectional area of the flow test section.

The data acquisition process is shown in Figure 5. The pressure values at both ends of the two samples were collected by three sensors, and the current signal was transmitted to the computer terminal through the DAQ card. After being processed by LabVIEW software (National Instruments, Austin, TX, USA), the pressure values at both ends of the samples can read directly on the operation panel, then the pressure drop and drag reduction rate were calculated.

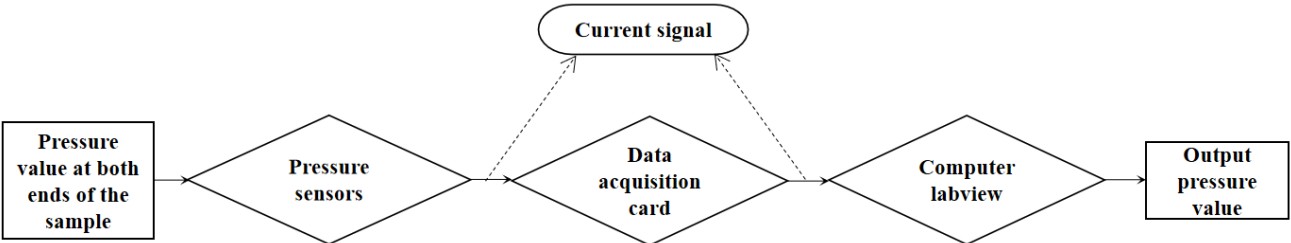

**Figure 5.** Schematic diagram of the pressure data acquisition.

In order to check the reliability of the setup, the pressure sensors and electromagnetic flow meter were calibrated. The calibrated current equation of three sensors are shown in Equations (1)–(3), respectively:

$$Y_0 = 26.637x_0 + 3.9377 \tag{1}$$

$$Y_1 = 26.902x_1 + 3.9649 \tag{2}$$

$$Y_2 = 26.637x_2 + 3.9377 \tag{3}$$

The calibration equation of the electromagnetic flow meter is Equation (4):

$$Y_3 = \frac{1}{0.424926}x_3 + 3.9612 \tag{4}$$

We input the slope and intercept of the four fitting curves into the settings of each path of LabVIEW panel to improve the accuracy. Slope and intercept represent the sensitivity and offset, respectively.

### 2.3.2. Experiment Plan

The samples of $70 \times 64 \times 5$ mm$^3$ size were placed in the channel. The flow velocity was adjusted, after the flow velocity stabilized, 5 s data were collected each time, then we take an average of the readings from each sensor. The pressure drop at both ends of the sample can be obtained from the difference in readings between adjacent pressure sensors. Pressure drop (*PD*) and drag reduction rate (*k*) were calculated by Equations (5) and (6), respectively:

$$PD = P_{\text{before}} - P_{\text{after}} \tag{5}$$

$$k = \frac{PD_{smooth} - PD_{bionic}}{PD_{smooth}} \times 100\% \tag{6}$$

where *PD* is the pressure drop, $P_{\text{before}}$ is the pressure value before the sample, $P_{\text{after}}$ is the pressure value after the sample, $PD_{smooth}$ is the pressure drop between two ends of the smooth sample, and $PD_{bionic}$ is the pressure drop between two ends of the bionic sample.

## 3. Results and Analysis
### 3.1. Structure of the Loach Scales

Figure 6 shows an image of *Paramisgurnus dabryanus* loach and its scales. There is some mucus on Figure 6a, and after removing the mucus (Figure 6b), the arrangement of scales can be clearly observed. The diameter of a single scale is 0.60 mm, the scales are in diamond arrangement with an array distance of 0.43 mm.

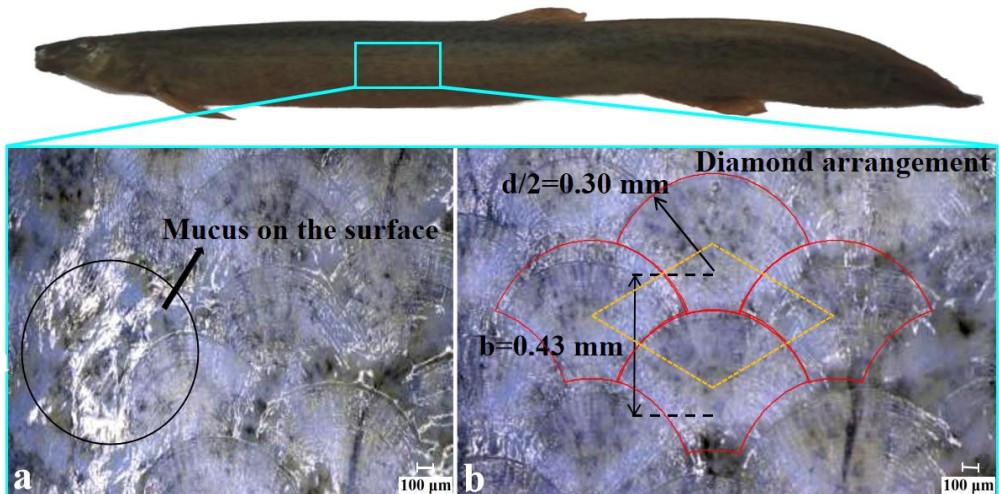

**Figure 6.** Arrangement of the loach scales. (**a**) Mucus on the scales; (**b**) Size and arrangement of scales after removing the mucus.

### 3.2. Results of the Channel Test

The drag reduction rate can be obtained by comparing the pressure drop between two ends of the smooth and bionic samples under a certain flow velocity. The results are shown in Table 2. The number of the pressure sensor fluctuates in the test process, so the average value of three flow channel tests is taken to get more scientific test results. Although the electromagnetic flow meter and pressure sensor have been strictly calibrated before use, there are still errors in the self-built flow channel to a certain extent. Moreover, the flow state during the test is turbulent, so there are some differences in the values of each test. However, it can be seen from the results that the overall trend presents a drag reduction effect, and the relative standard deviation (RSD) is within 5%, so the test results are credible.

**Table 2.** Channel test results.

| Volume Flow Rate (m³/h) | PD Smooth (Pa) | | | | SD | RSD |
| --- | --- | --- | --- | --- | --- | --- |
| | Smooth Test 1 | Smooth Test 2 | Smooth Test 3 | Average Value | | |
| 0.204 | 672.5468 | 683.2486 | 711.3469 | 689.0474 | 16.3622 | 2.37% |
| 0.303 | 665.9226 | 669.5486 | 703.6699 | 679.7137 | 17.0042 | 2.50% |
| 0.398 | 680.7184 | 650.3495 | 625.5408 | 652.2029 | 22.5643 | 3.46% |
| 0.507 | 646.1646 | 625.1648 | 629.6532 | 633.6608 | 9.0293 | 1.42% |
| 0.611 | 659.2171 | 665.3649 | 721.5431 | 682.0417 | 28.0442 | 4.11% |
| 0.715 | 639.2333 | 645.3654 | 702.5038 | 662.3675 | 28.4909 | 4.30% |
| 0.792 | 642.6022 | 659.2654 | 643.2336 | 648.3671 | 7.7106 | 1.19% |
| 0.900 | 636.4690 | 642.5895 | 687.5243 | 655.5276 | 22.7626 | 3.47% |
| 1.06 | 691.1193 | 746.3654 | 757.8527 | 731.7791 | 29.1308 | 3.98% |
| Volume Flow Rate (m³/h) | PD Bionic (Pa) | | | | SD | RSD |
| | Bionic Test 1 | Bionic Test 2 | Bionic Test 3 | Average Value | | |
| 0.204 | 628.4862 | 642.4862 | 664.9267 | 645.2997 | 15.0092 | 2.33% |
| 0.303 | 647.1653 | 689.5464 | 641.1235 | 659.2784 | 21.5444 | 3.27% |
| 0.398 | 608.2772 | 598.4687 | 560.1497 | 588.9652 | 20.7654 | 3.53% |
| 0.507 | 527.4023 | 547.6554 | 553.1338 | 542.7305 | 11.0670 | 2.04% |
| 0.611 | 560.2364 | 543.6548 | 520.4762 | 541.4558 | 16.3063 | 3.01% |
| 0.715 | 499.4779 | 489.3478 | 538.4779 | 509.1012 | 21.1801 | 4.16% |
| 0.792 | 643.6147 | 638.4715 | 707.0017 | 663.0293 | 31.1640 | 4.70% |
| 0.900 | 650.4684 | 615.2646 | 621.0669 | 628.9333 | 15.4108 | 2.45% |
| 1.06 | 710.8683 | 642.5698 | 678.2129 | 677.2170 | 27.8916 | 4.12% |

The drag reduction rate can be obtained through *PD* calculation. The results are shown in Table 3 below. It can be seen that when the flow velocity is 1.683 m/s, the drag reduction rate reaches the maximum 23%.

**Table 3.** Drag reduction rate at different flow velocities.

| Flow Velocity (m/s) | 0.48 | 0.713 | 0.937 | 1.193 | 1.438 | 1.683 | 1.864 | 2.118 | 2.498 |
|---|---|---|---|---|---|---|---|---|---|
| $k$ (%) | 6% | 3% | 10% | 14% | 21% | 23% | −2% | 4% | 7% |

### 3.3. Numerical Simulation

#### 3.3.1. Establishment of the Model

Using the loach scales as bionic prototype, a model with scales as units was established. The geometric parameters are the same as the artificial sample, namely, h is 20 µm, d is 0.60 mm, a is 0.36 mm, and b is 0.43 mm (Figure 7).

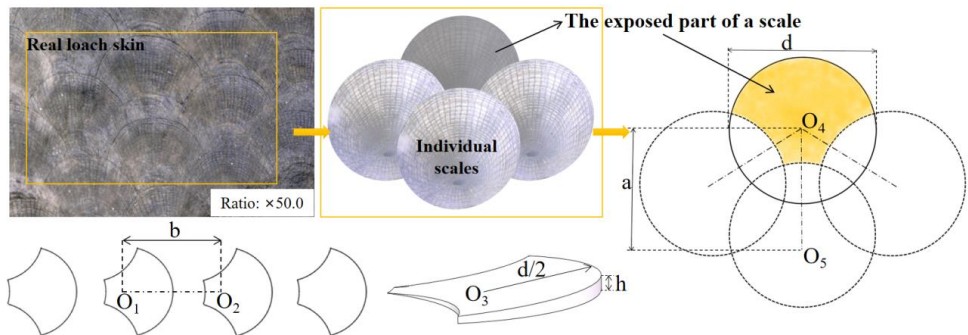

**Figure 7.** Structure unit parameters.

#### 3.3.2. Setting of Initial Conditions

The sample in this test is aluminum with a rigid surface and no obvious deformation under the fluid. In order to calculate efficiently and quickly, fluid-structure interaction is not considered and computational fluid dynamics (CFD) simulation calculations are selected. A cuboid calculation domain was built with a size of $30 \times 8 \times 6$ mm$^3$ (Figure 8), with the inlet on the left and the outlet on the right. We chose the method "Magnitude, normal to Boudary" to define the inlet velocity and it is a uniform flow profile. The bottom surface containing the bionic structure is called the B-wall, the smooth surface on the top is the S-wall. The test zone in red color in Figure 8 is $6 \times 8$ mm$^2$ and the size parameters of the scale element are exactly the same as those of the experiment sample. The other walls are all named Wall, which are set as no sliding wall surfaces.

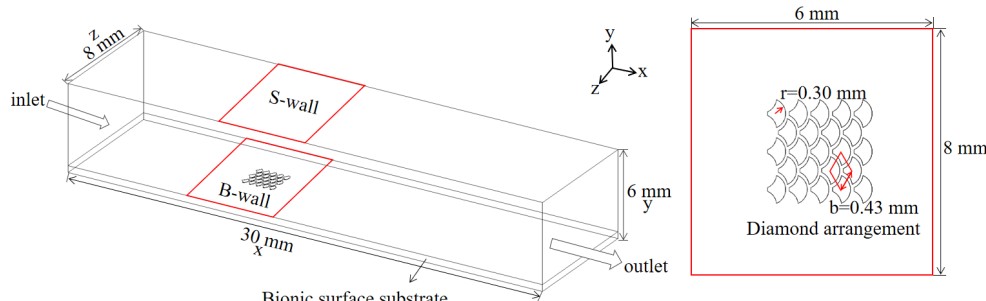

**Figure 8.** Computational domain.

The density of aluminum is 2719 kg/m$^3$. The flow field medium was water, whose density was 998.2 kg/m$^3$. The dynamic viscosity coefficient was 0.001003 Pa·s and the

time was set to transient. The calculation time step size was 0.001 s, the number of time steps was 1000, and the number of iterations was set to 10. The hydraulic diameter and turbulence intensity were used to make a preliminary definition of the flow field and calculated by Equations (7) and (8):

$$R_{\text{h}} = \frac{2zy}{z+y} \tag{7}$$

$$I = \frac{0.16}{R_e^{\frac{1}{8}}} \tag{8}$$

where $R_{\text{h}}$ is the hydraulic diameter, which is the equivalent diameter corresponding to the circular pipe. $y$ and $z$ are the length and width of the inlet of the flow field area, respectively. $I$ is the intensity of turbulence and $R_e$ is the Reynolds number. In this paper, the $R_e$ under flow conditions are all greater than those under transition conditions, so the RNG k-$\varepsilon$ turbulence model was selected for calculation.

### 3.3.3. Mesh Generation

The mesh module of ANSYS workbench was selected to generate the mesh. Considering the existence of sharp angles on the bionic surface, the hexahedral structured mesh and tetrahedral unstructured mesh were used to generate the mesh. After repeated attempts, the mesh size was finally set to be 0.25 mm in the middle part of the fluid region and 0.1 mm in the boundary layer region. Due to the small size of fish scale unit, the finest meshes meshes were generated on the bionic surface with a size of 0.025 mm. Figure 9 shows the mesh image of the whole computational region and local bionic surface.

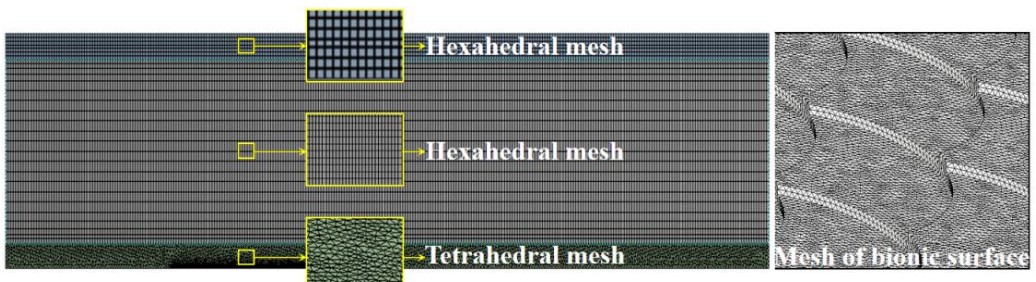

**Figure 9.** Mesh image of the whole computational region and local bionic surface.

### 3.3.4. Simulation Scheme

In the simulation, both the smooth and bionic surface were analyzed under exactly the same calculated area and flow velocity range of 0.8–13 m/s. Compared with the smooth surface, the drag reduction rate ($\eta$) of the bionic surface can be obtained through Equation (9):

$$\eta = \frac{F_{smooth} - F_{bionic}}{F_{smooth}} \times 100\% \tag{9}$$

where, $F_{smooth}$ is the total resistance on the smooth surface, $F_{bionic}$ is the total resistance on the bionic surface, and $\eta$ is the drag reduction rate. When an object is in contact with a fluid and in relative motion, the total resistance is formed by differential pressure resistance and viscous resistance (Equation (10)):

$$F = F_{\text{dp}} + F_{\text{v}} \tag{10}$$

where, $F$ is the total resistance, $F_{dp}$ is the differential pressure resistance and $F_v$ is the viscous resistance. The main influencing factors on differential pressure resistance include cross-sectional area, the shape of the object and posture in the flow field. The viscous resistance is related to the roughness, the relative speed, and the friction coefficient [10].

### 3.3.5. Results of Numerical Simulation

Table 4 shows the resistance value of both surfaces under conditions of low, medium and high speed, as well as the drag reduction rate of viscous resistance and the total resistance.

**Table 4.** Drag reduction rate under different flow velocities.

| Flow Velocity (m/s) | | Re | Resistance of B-Wall (N) | | | Resistance of S-Wall (N) | | | Drag Reduction Rate | |
|---|---|---|---|---|---|---|---|---|---|---|
| | | | Differential Pressure | Viscous | Total | Differential Pressure | Viscous | Total | Viscous Resistance | Total Resistance |
| Low | 0.8 | 5485.6 | $1.73 \times 10^{-6}$ | $2.30 \times 10^{-4}$ | $2.32 \times 10^{-4}$ | 0 | $2.41 \times 10^{-4}$ | $2.41 \times 10^{-4}$ | 4.41% | 3.70% |
| | 1 | 6857.0 | $2.66 \times 10^{-6}$ | $3.36 \times 10^{-4}$ | $3.38 \times 10^{-4}$ | 0 | $3.56 \times 10^{-4}$ | $3.56 \times 10^{-4}$ | 5.57% | 5.00% |
| | 1.2 | 8228.4 | $4.79 \times 10^{-6}$ | $4.57 \times 10^{-4}$ | $4.62 \times 10^{-4}$ | 0 | $4.94 \times 10^{-4}$ | $4.94 \times 10^{-4}$ | 7.51% | 6.54% |
| | 1.4 | 9599.8 | $5.28 \times 10^{-6}$ | $5.95 \times 10^{-4}$ | $6.00 \times 10^{-4}$ | 0 | $6.50 \times 10^{-4}$ | $6.50 \times 10^{-4}$ | 8.46% | 7.65% |
| | 1.6 | 10971.2 | $6.99 \times 10^{-6}$ | $7.46 \times 10^{-4}$ | $7.53 \times 10^{-4}$ | 0 | $8.10 \times 10^{-4}$ | $8.10 \times 10^{-4}$ | 7.87% | 7.01% |
| | 1.8 | 12342.6 | $8.99 \times 10^{-6}$ | $9.11 \times 10^{-4}$ | $9.20 \times 10^{-4}$ | 0 | $9.76 \times 10^{-4}$ | $9.76 \times 10^{-4}$ | 6.63% | 5.71% |
| Mid | 2 | 13714.0 | $1.13 \times 10^{-5}$ | $1.09 \times 10^{-3}$ | $1.10 \times 10^{-3}$ | 0 | $1.16 \times 10^{-3}$ | $1.16 \times 10^{-3}$ | 5.93% | 4.95% |
| | 2.2 | 15085.4 | $1.40 \times 10^{-5}$ | $1.28 \times 10^{-3}$ | $1.30 \times 10^{-3}$ | 0 | $1.36 \times 10^{-3}$ | $1.36 \times 10^{-3}$ | 5.79% | 4.77% |
| | 2.4 | 16456.8 | $1.70 \times 10^{-5}$ | $1.49 \times 10^{-3}$ | $1.51 \times 10^{-3}$ | 0 | $1.58 \times 10^{-3}$ | $1.58 \times 10^{-3}$ | 5.94% | 4.86% |
| | 2.6 | 17828.2 | $2.04 \times 10^{-5}$ | $1.71 \times 10^{-3}$ | $1.73 \times 10^{-3}$ | 0 | $1.82 \times 10^{-3}$ | $1.82 \times 10^{-3}$ | 6.13% | 5.02% |
| | 2.8 | 19199.6 | $2.41 \times 10^{-5}$ | $1.94 \times 10^{-3}$ | $1.97 \times 10^{-3}$ | 0 | $2.07 \times 10^{-3}$ | $2.07 \times 10^{-3}$ | 6.27% | 5.11% |
| | 3 | 20571.0 | $2.83 \times 10^{-5}$ | $2.19 \times 10^{-3}$ | $2.22 \times 10^{-3}$ | 0 | $2.34 \times 10^{-3}$ | $2.34 \times 10^{-3}$ | 6.31% | 5.10% |
| High | 4 | 27428.0 | $5.59 \times 10^{-5}$ | $3.63 \times 10^{-3}$ | $3.68 \times 10^{-3}$ | 0 | $3.83 \times 10^{-3}$ | $3.83 \times 10^{-3}$ | 5.28% | 3.82% |
| | 5 | 34285.0 | $9.63 \times 10^{-5}$ | $5.40 \times 10^{-3}$ | $5.49 \times 10^{-3}$ | 0 | $5.65 \times 10^{-3}$ | $5.65 \times 10^{-3}$ | 4.43% | 2.73% |
| | 7 | 47999.0 | $2.24 \times 10^{-4}$ | $9.93 \times 10^{-3}$ | $1.02 \times 10^{-2}$ | 0 | $1.04 \times 10^{-2}$ | $1.04 \times 10^{-2}$ | 4.53% | 2.39% |
| | 9 | 61713.0 | $4.29 \times 10^{-4}$ | $1.59 \times 10^{-2}$ | $1.63 \times 10^{-2}$ | 0 | $1.66 \times 10^{-2}$ | $1.66 \times 10^{-2}$ | 4.14% | 1.55% |
| | 11 | 75427.0 | $7.29 \times 10^{-4}$ | $2.32 \times 10^{-2}$ | $2.40 \times 10^{-2}$ | 0 | $2.40 \times 10^{-2}$ | $2.40 \times 10^{-2}$ | 3.17% | 0.13% |
| | 13 | 89141.0 | $1.14 \times 10^{-3}$ | $3.20 \times 10^{-2}$ | $3.31 \times 10^{-2}$ | 0 | $3.26 \times 10^{-2}$ | $3.26 \times 10^{-2}$ | 1.84% | −1.65% |

It can be seen from Table 4, the total resistance is formed by the differential pressure resistance and the viscous resistance, and the viscous resistance accounts for a larger proportion in the total resistance. The differential pressure resistance on the bionic wall increases by almost three orders of magnitude as the Re is increased, whereas that on the smooth wall remains unchanged at zero. Although the pressure differential resistance of the bionic surface is greater than that of the smooth surface, its viscous resistance is far less than that of the smooth surface. Therefore, the overall trend of drag reduction is obvious.

Figure 10a shows the total resistance of both surfaces among the whole range of flow velocity. Figure 10b,c show the detailed information of the low-speed group and the medium-speed group, respectively. In the low-speed group, the total resistance on the smooth surface was higher than that on the bionic surface and it increased rapidly. In the medium-speed group, the total resistance of the both surfaces increased simultaneously, and the drag reduction rate fluctuated at about 5%. In the high-speed group, the total resistance of the both surfaces increased rapidly and became equal gradually. When the flow velocity was 13 m/s, the resistance on the bionic sample was more than that on the smooth sample.

When the sample moves in the fluid, the viscous resistance accounts for a large proportion of the total resistance, so the variation trend of the viscous resistance affects the variation trend of the total resistance to a great extent. The relation curve of flow velocity and drag reduction rate is shown in Figure 11. At 1.4 m/s, the total drag reduction rate reached a maximum value of 7.65%. With the increase of flow velocity, the drag reduction rate gradually decreased. The variation trend of viscous drag reduction rate was roughly the same as that of total drag reduction rate. With the increase of flow velocity, the proportion of differential pressure resistance in total drag increased gradually, so the drag reduction effect was lost to some extent.

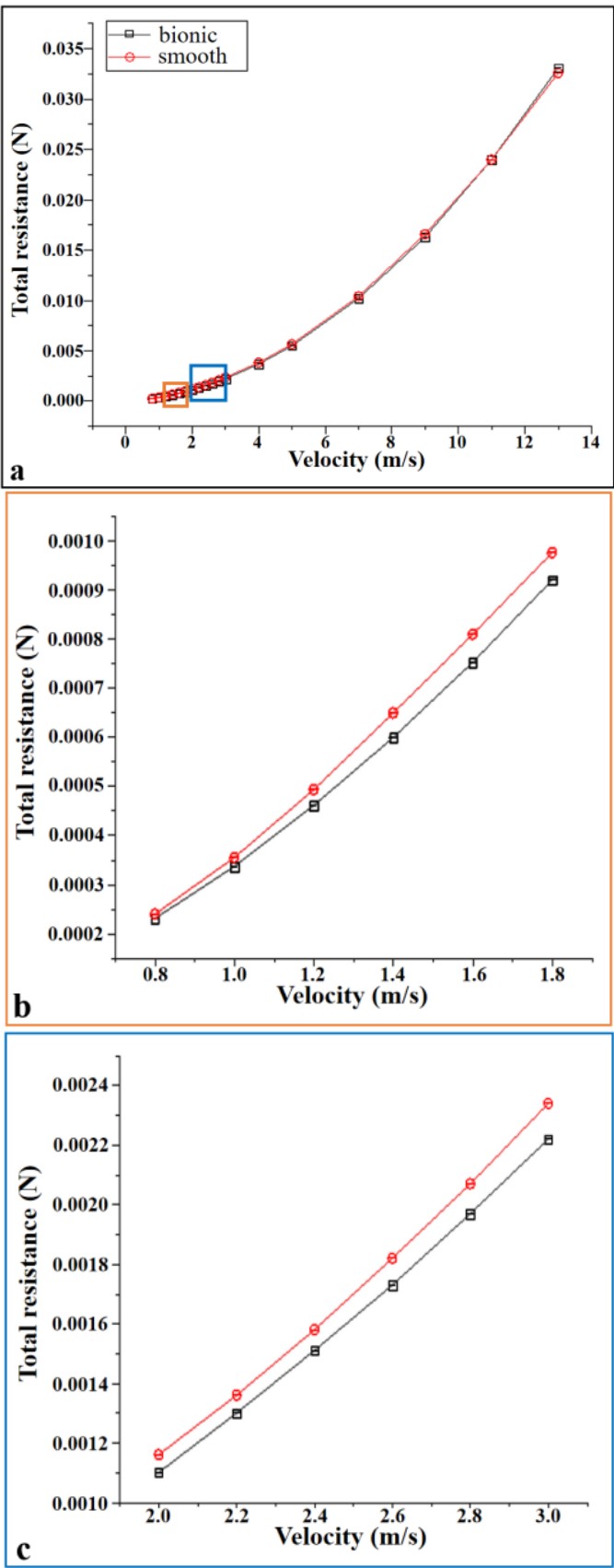

**Figure 10.** Flow velocity-resistance diagram. (**a**) The total resistance of both surfaces among the whole range of flow velocity. (**b**) The detailed information of the low-speed group; (**c**) The detailed information of the medium-speed group.

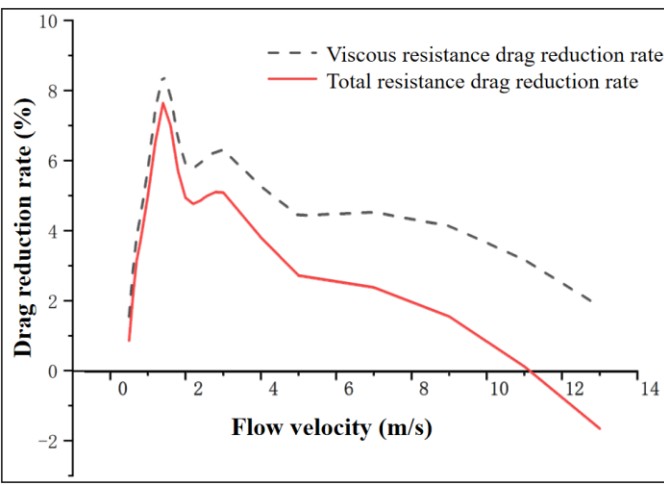

**Figure 11.** The relation curve of flow velocity and drag reduction rate.

*3.4. Comparison between Results of Numerical Simulation and Experiment*

The accuracy of the experimental result was verified by CFD. Figure 12 shows the comparison of drag reduction rate between experiment and numerical simulation.

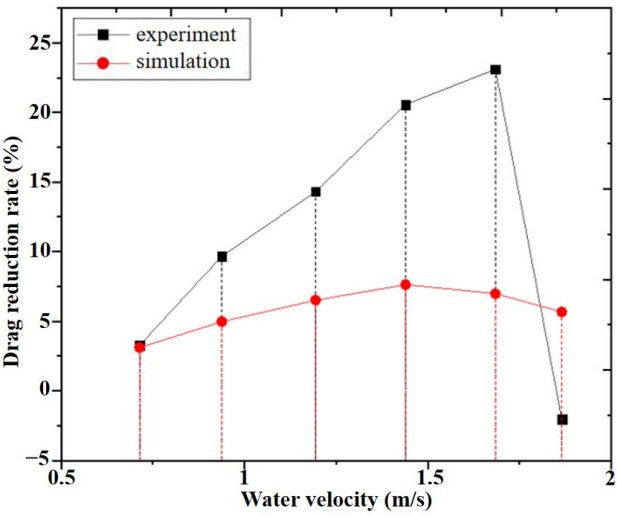

**Figure 12.** Comparison of experiment and numerical simulation results.

As can be seen from the Figure 12, both the experimental and numerical simulation results showed the drag reduction effect of the bionic sample. Within the water velocity range of 0.5–2 m/s, the peak drag reduction rate obtained in the experiment was 23%, which occurred at the velocity of 1.68 m/s. In the numerical simulation, the peak drag reduction rate was 7.65%, which occurred at the velocity of 1.4 m/s. As for the difference of peak drag reduction rate, due to the small calculation area of numerical simulation and the relatively large size of test samples in the actual experiment, the difference between them was hundreds of times. In addition, the processing accuracy of test samples would also affect the accuracy of test results.

## 4. Mechanism Analysis

Figure 13 shows the velocity cloud diagram of the x-y cross-section. Figure 13a,b show detailed views of the smooth surface and the bionic surface, respectively. It can be seen that the contours of the bionic surface are smoother than the smooth surface. Therefore, the structure of biomimetic fish scale arrangement can effectively control the flow near the wall and reduce energy loss.

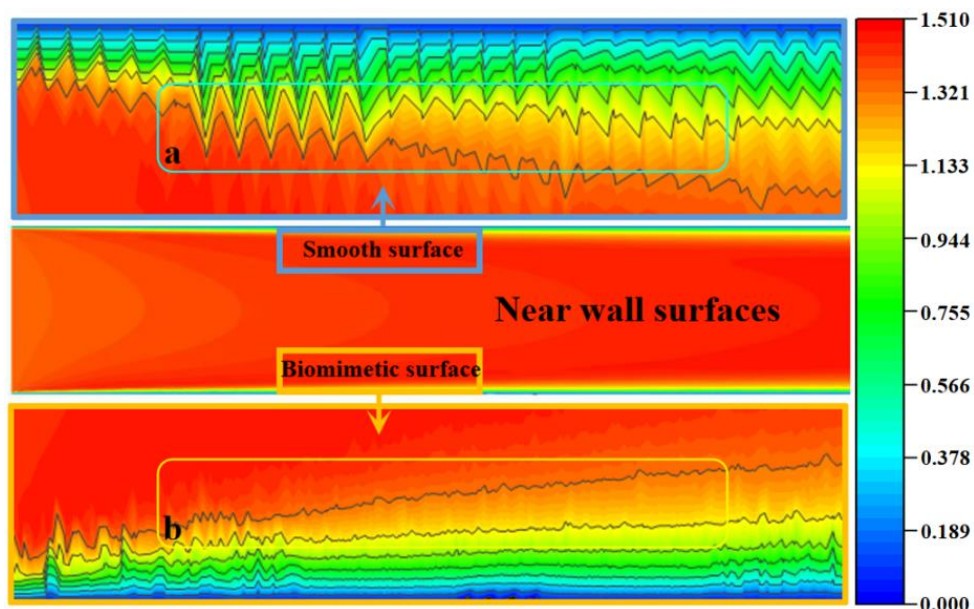

**Figure 13.** Velocity cloud diagram of the x-y cross-section. (**a**) Detailed view of the smooth surface; (**b**) Detailed view of the bionic surface.

The velocity distribution was analyzed at the three positions near the wall of the x-y plane, as shown in Figure 14. It can be seen that in the case of the same distance in the y direction, the speed of the bionic surface increases slowly and the speed gradient and viscous resistance is small, so the bionic surface can increase the thickness of boundary layer and reduce the resistance.

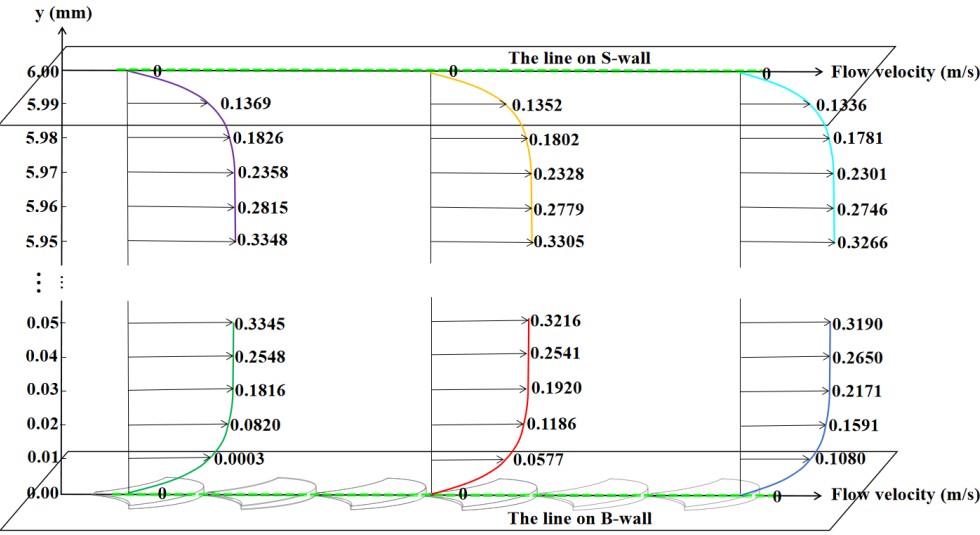

**Figure 14.** Velocity distribution near the wall.

Figure 15 shows the shear force cloud diagram and pressure cloud diagram of the upper and lower surfaces. By extracting the values on the center lines of the test area in Figure 15a,b and Figure 16a,b were obtained.

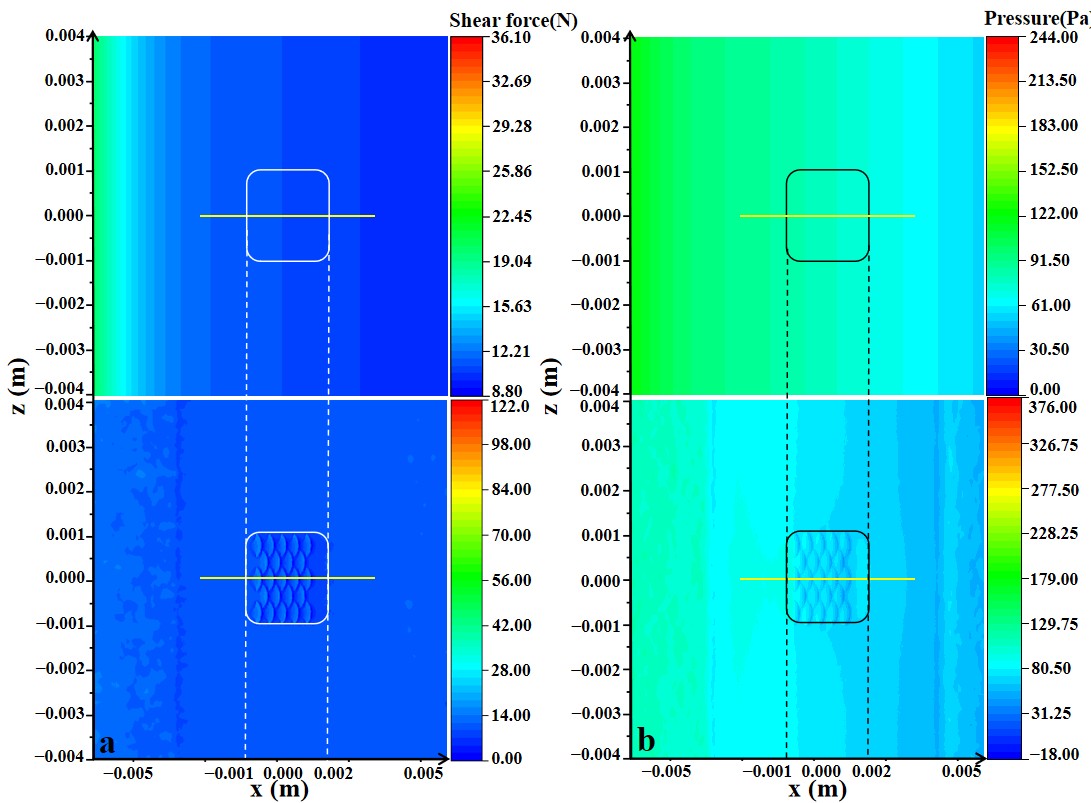

**Figure 15.** Shear force and pressure contour of the upper and lower surfaces. (**a**) Shear force contour; (**b**) Pressure contour.

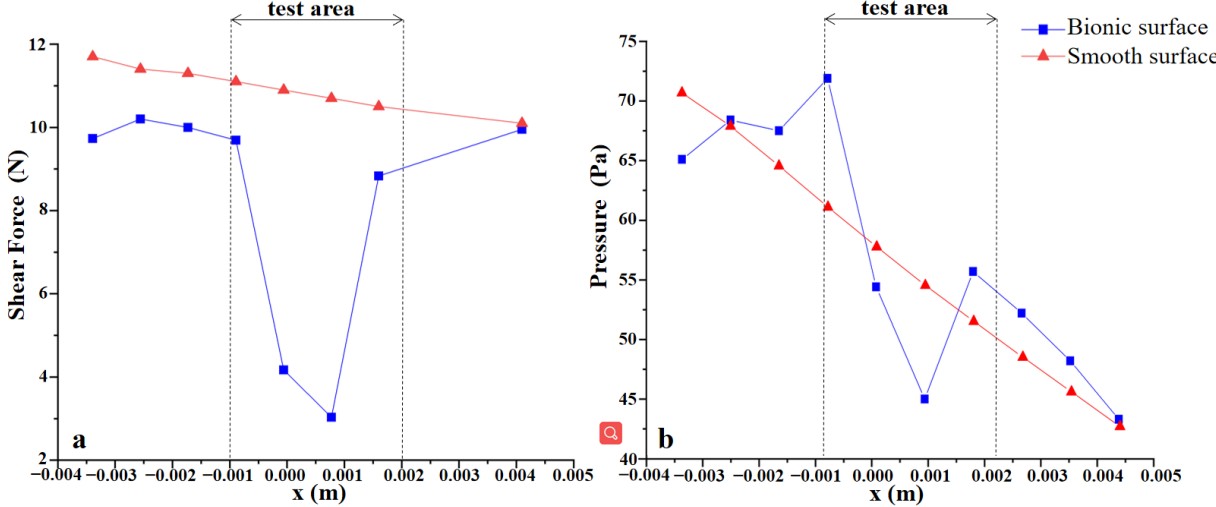

**Figure 16.** Shear force and pressure diagram of the center line of the test area. (**a**) Shear force line diagram; (**b**) Pressure line diagram.

The shear force and pressure on the bionic surface decreased rapidly in reaching the test area, and the corresponding x-coordinate was −0.001 m. After passing through the bionic region, they reached the minimum value when the x-coordinate was about 0.001 m. After water flowed through the fish scale array, the wall shear force increased until it was almost equal to the wall shear force at the same position as the smooth surface. As can be seen from the pressure cloud diagram in Figure 15b, there was a high-pressure backflow area at the back of the array scales, so the pressure tended to rise when the x-coordinate was between 0.001 m and 0.002 m. After passing through this region, the pressure decreased again to almost the same value at the same coordinates on the smooth surface.

In Figure 17, the yellow frame is the area where the bionic sample is placed, and the red one for smooth sample. As can be seen from the detailed views, the bionic sample area approached the low-pressure area of 75 Pa pressure isoline earlier, and with the increase of the scale number, the area of the low-pressure area became larger and larger. The presence of a low pressure region will absorb a certain volume of fluid. The reasonable distribution, size and shape of these scales connect each low-pressure water trapping area together to form a stable and continuous liquid film, which plays a role of fluid lubrication instead of solid-liquid contact to achieve the purpose of drag reduction.

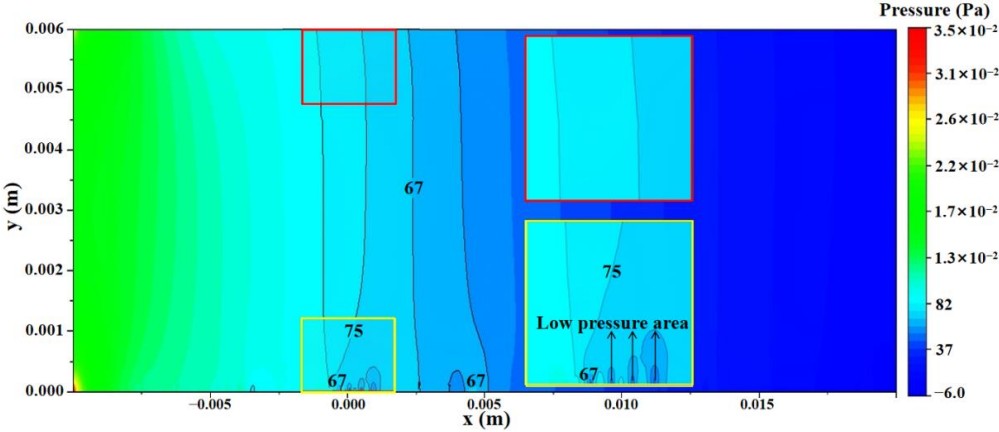

**Figure 17.** Pressure cloud diagram of the x-y cross-section.

Figure 18 shows the distribution of turbulent kinetic energy. It can be seen that the turbulent kinetic energy at all positions of the smooth surface is higher than that of the bionic surface. For example, the maximum and minimum values of turbulent kinetic energy of smooth surface are 0.077 and 0.011, respectively, while that corresponding to the bionic surface are 0.066 and 0, respectively. On the other hand, the turbulent kinetic energy of the bionic surface is more evenly distributed than that of the smooth surface in the test area.

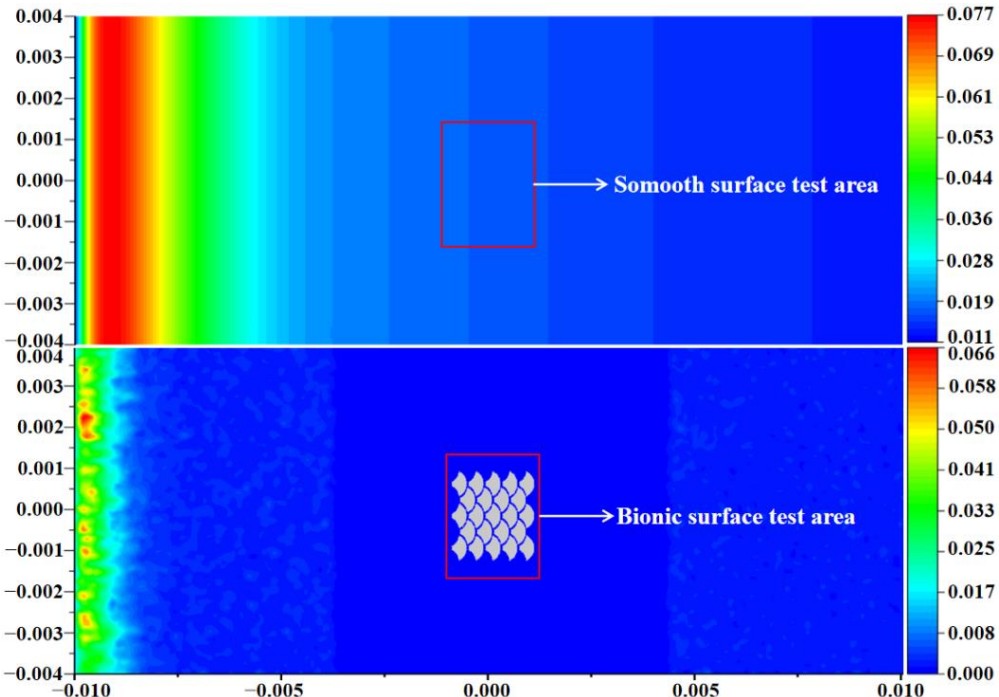

**Figure 18.** Turbulent energy cloud diagram on the upper and lower surfaces.

### 4.1. X-z Plane "Corrugated" Flow

Figure 19 shows the distribution of pressure and velocity along the array direction. Figure 19a shows the location and direction of value line. Along the red, green and blue lines, the variation of pressure and velocity are shown in Figure 19b,c. It can be seen that the high and low pressure and high and low velocity regions appeared periodically. As in the flow direction, and the values of both pressure and velocity decreased gradually. The pressure difference between inlet and outlet was 44.74 Pa after calculation, and the velocity difference was 0.13 m/s.

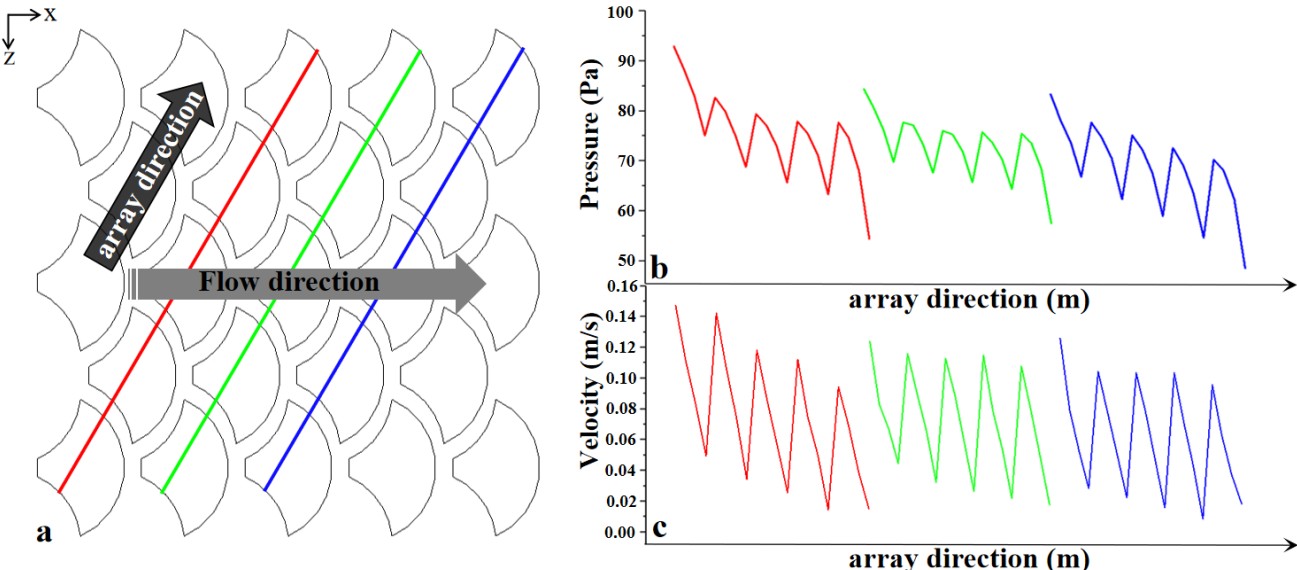

**Figure 19.** Distribution of pressure and velocity along array direction. (**a**) Value location and direction selection; (**b**) Distribution of pressure; (**c**) Distribution of velocity.

Figure 20 shows the wavy flow on the surface. As shown in Figure 20a,b the direction of streamlines in overlapping regions of fish scales showed regular periodic changes, the streamlines appear to be "corrugated" within the overlapping area marked with yellow. In the blue square of Figure 20b, due to the change in pressure, the streamlines dispersed from the middle to the outskirts firstly, and then gathered from the outside to the inside. The detailed view in the orange square is shown in Figure 20c, and the direction of streamlines changed between every two scales. Muthuramalingam [35] also found that the overlapping arrangement of bionic fish scales could produce streaks along the flow. The fish scale array can stabilize the laminar boundary layer and delay the transformation. The drag was reduced consequently.

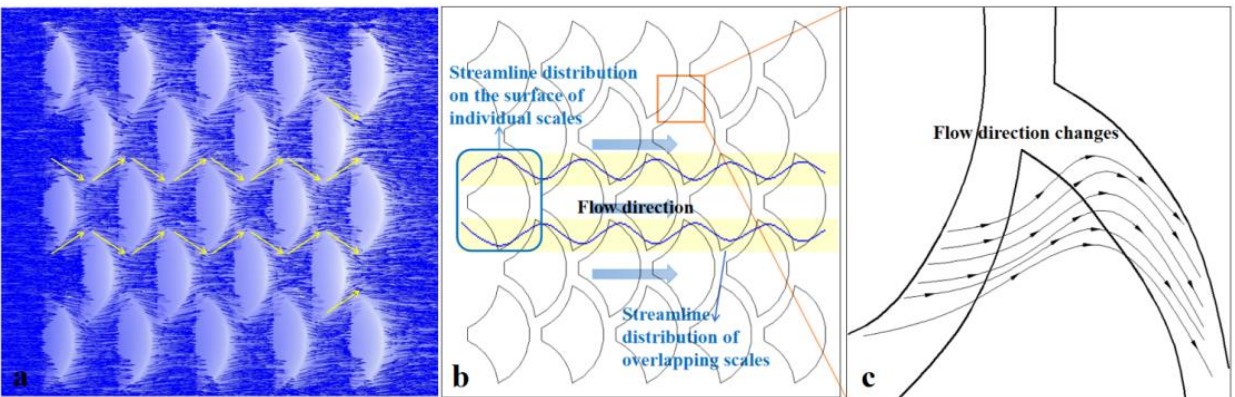

**Figure 20.** Wavy flow state diagram. (**a**) vVelocity vector diagram of the scale array region; (**b**) The "corrugated" streamlines; (**c**) The direction of streamlines changed between every two scales.

### 4.2. Stable High and Low Speed Streaks and Variation Trend of Bionic Surface Velocity

Figure 21a shows the average velocity distribution of each region obtained in this experiment. It can be seen that only two velocity peaks appear in the middle position of fish scales in the second and fourth rows. Figure 21b shows the experimental results of Muthuramalingam's team. They also set up five rows of fish scale arrays along the flow direction, and the test results showed that the high-speed streaks appeared four times.

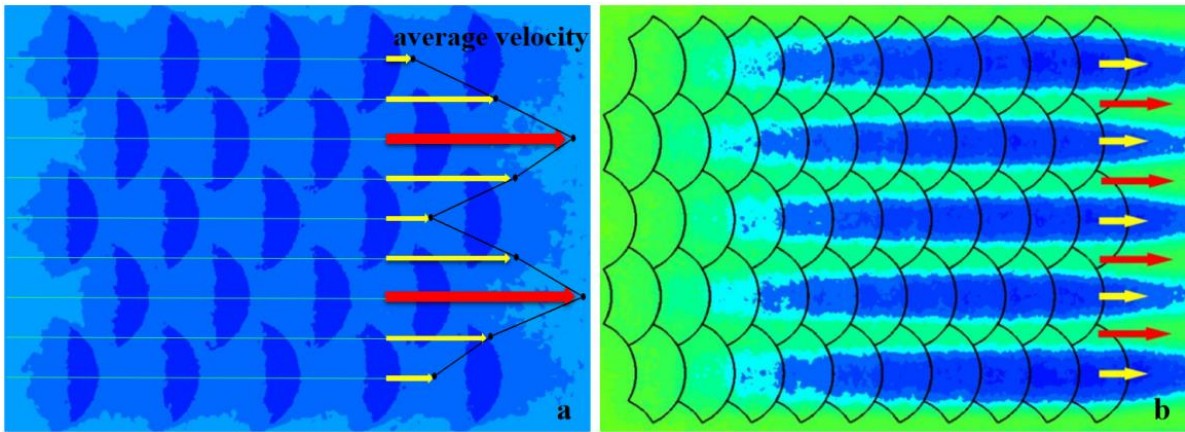

**Figure 21.** Velocity distribution on the bionic surface. (**a**) The result of this research; (**b**) The result of Muthuramalingam, M.'s research [36]; Copyright 2019 Journal of Experimental Biology.

There are two main reasons for this situation. In this paper, the size of scale is smaller and the distance between each column is larger contrast to the scale size, so the high-speed streaks on both sides of the second and fourth row develops and merges into one peak. In the reference [35], due to the large size of scale and the distance between the columns is smaller contrast to the scale size, the peak velocities can occur in adjacent overlapping regions at the same time.

A conclusion that can be drawn from the above is that the periodic change of velocity in z direction could be influenced by scale size and distribution. Figure 22 shows the velocity distribution of each column of fish scales. Nine positions are taken in the array area to analyze the velocity data on each line. In the z-axis direction, the average velocity of each scale column is shown in Figure 22a. The nine points on the red solid line correspond to the average speed of each column. We take 20 points with equal spacing on each scale column and calculate the velocity of each point to form a broken line chart, as shown in Figure 22b. As can be seen from the figure, from the entrance to the exit, the peak velocity of columns 1–9 gradually decreases and tends to be stable.

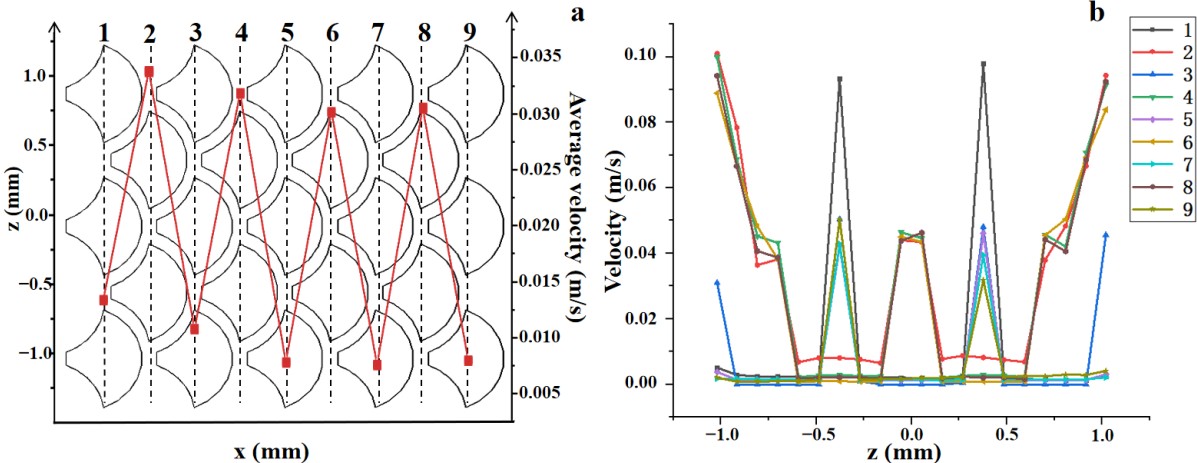

**Figure 22.** Velocity distribution of each scale column. (**a**) Average velocity on each of scale column; (**b**) The velocity distribution of each column.

As can be seen from the definition of Reynolds number, when the flow medium and flow environment remain unchanged, the Reynolds number decreases when the flow velocity decreases. 15 points were selected at the same position of the smooth surface and the bionic surface like Figure 23a. The velocity change of each points was shown in Figure 23b. It can be seen that after passing the test area (yellow frame) in Figure 23a, the velocity decrease value of the bionic surface was $L_{B-wall}$ = 0.05 m/s, and that of the smooth surface was $L_{S-wall}$ = 0.025 m/s. Namely, the Reynolds number of bionic surface decreases more.

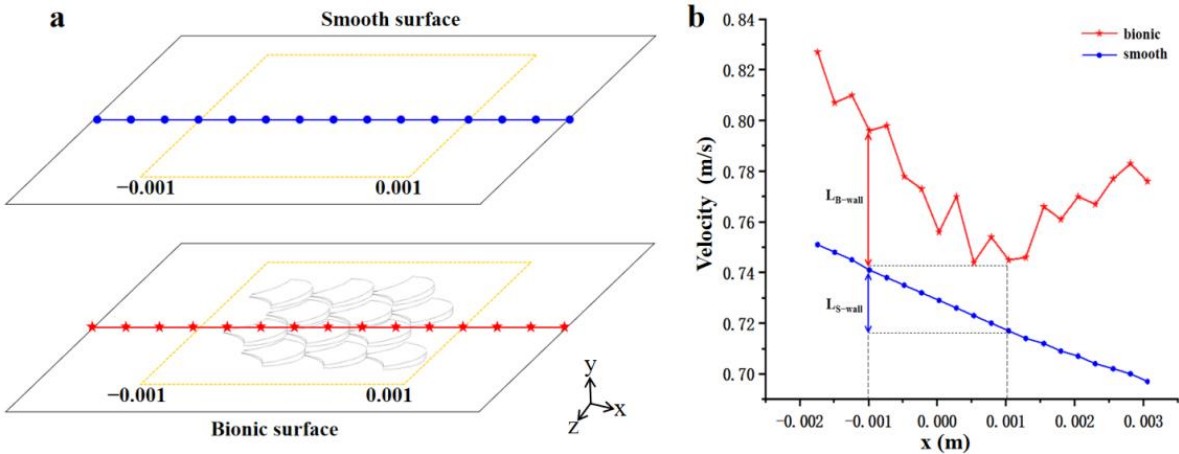

**Figure 23.** The change of velocity in the flow direction of smooth surface and bionic surface. (**a**) The position of the value line; (**b**) A diagram of velocity variation along the direction of flow.

### 4.3. The Vortex Lifting Mechanism in the x-y Plane

Figure 24 shows the distribution of vortices and the streamline of the x-y cross section. During the flow process, a clockwise vortex was generated behind each scale. The fluid was lifted through the scale and reached the highest point at the end edge of the scale. The vortex behind the scale makes the high-speed fluid move away from the near-wall region, which effectively reduces the near-wall velocity and greatly weakens the turbulence. In addition, these vortices are located in low-pressure regions that continuously attract fluid, converting solid-liquid friction into liquid-liquid friction and effectively reducing drag [37]. The part marked with yellow in Figure 24c is the schematic diagram of the contact surface between liquid and a solid. Due to the presence of a thin film of liquid on the low-pressure surface, bionic surface can reduce the contact area and reduce the frictional resistance [38,39]. Therefore, the hypothesis that surface waves can reduce the resistance was reasonable and effective [40,41].

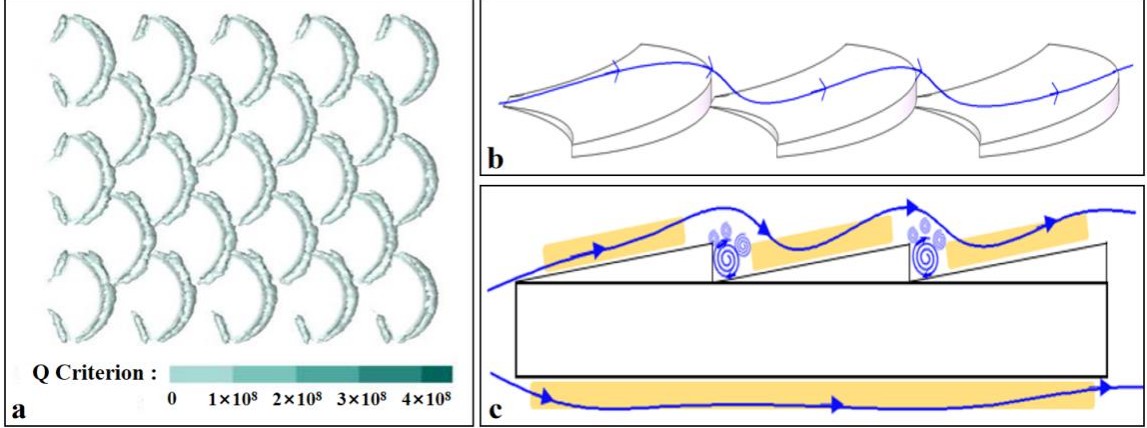

**Figure 24.** The vortices distribution and the streamline of the x-y cross section. (**a**) Vortices diagram generated according to the Q criterion; (**b**) The streamline on the surface; (**c**) Schematic diagram of the contact surface between liquid and a solid.

In this paper we did not examine the influence of height and surface features on drag reduction. The work of modeling, sample preparation and simulation analysis were carried out according to the structural parameters of the original loach skin prototype. Subsequent studies will be carried out on the influence of different structural parameters on drag reduction.

## 5. Conclusions

In order to reduce the underwater flow resistance, inspired by the structure and distribution of the skin of the *Paramisgurnus dabryanus* loach, a bionic non-smooth drag reduction surface was studied. Based on experimental and numerical simulation, the following conclusions were drawn:

(1) An aluminum sample was processed by the CNC method, and a flow channel was built. The sample size was $70 \times 64 \times 5$ mm$^3$ with hundreds of bionic fish scale units. The drag reduction rate of the sample could reach 23% when the velocity was 1.683 m/s.

(2) A model with 23 scales was built for numerical simulation. A drag reduction effect was achieved within a velocity range of 0.8–11 m/s, and the maximum drag reduction rate was 7.65% when the flow velocity was 1.4 m/s.

(3) The bionic micro-structure can control near-wall flow, reduce near-wall velocity gradient, increase boundary layer thickness, and delay the transition of layer flow turbulence.

(4) Due to the existence of high and low pressure zones, alternating high and low speed streaks were generated in the x direction. As the water flows over the bionic surface, it slows down significantly, which could effectively reduce the Reynolds number and delay the transition of laminar to turbulent flow.

(5) Vortices were generated behind each scale, which helped to form the liquid-liquid friction film and thus reducing the resistance.

**Author Contributions:** Methodology, L.W. and C.L.; software, J.W.; validation, J.W., G.L. and S.W.; investigation, J.Q.; resources, L.W.; data curation, J.W.; writing—original draft preparation, J.W.; writing—review and editing, L.W. and J.W.; supervision, X.F. All authors have read and agreed to the published version of the manuscript.

**Funding:** This research was funded by the National Natural Science Foundation of China (No. 51305282), Key Research Plan of Liaoning, China (No. 2020JH2/10700001), Project of Scientific Research Funding of Liaoning, China (No. LSNJC201908).

**Institutional Review Board Statement:** The study was conducted according to the guidelines of the Declaration of Helsinki, and approved by the Institutional Review Board (or Ethics Committee) of Shenyang Agricultural University. (protocol code 202108001 and date of approval 2021.3).

**Informed Consent Statement:** Not applicable.

**Data Availability Statement:** The data presented in this study is available within the article.

**Conflicts of Interest:** The authors declare no conflict of interest.

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
