# Peer review of "Study on the Drag Reduction Characteristics of the Surface Morphology of Paramisgurnus dabryanus Loach"

_coatings, doi:10.3390/coatings11111357_

Round 1
Reviewer 1 Report
This paper reveals the drag reduction characteristics of Paramisagurnus abryanus loach experimentally and numerically. I think the content is original and worthy of publication in the journal, but the following points need to be corrected. 1. There is a lack of description regarding experimental equipment, methods, and measurement accuracy. What are the types of pressure gauges and flow meter? Manufacturer? Range? Accuracy? 2. The size of the flow channel should be shown. 3. It should be indicated by the bulk Re number rather than the flow rate. Is the experiment laminar or turbulent? 4. Fig. 6: Why is the result of 1 not smooth? Also, it is necessary to consider the digits of the numbers on the horizontal axis. 5. Fig. 7: I don't understand the meaning of this experiment. What is the relationship with numerical simulation? 6. How does it compare with the result of the conventional flat plate? Or how does it compare to the result of the duct? This is necessary to prove that the experimental results are correct. 7. Express the scale of the fine structure with a wall unit. This comparison is necessary to consider the mechanism. 8. I can't find the result of the reduction of Reynolds stress. 9. Fig. 24: The plot should be increased a little more. Also, a velocity distribution perpendicular to the wall is required.
Reviewer 2 Report
The authors did a full investigation of the drag reduction rate of a water flow on bioinspired bionic surfaces (with a shape similar to P. abrayanus loach skin). They explored the topic from both an experimental and numerical point of view. The methodology is consistent, experiments and simulations were described and results were rationally explained.
I agree to accept this article after my minor comments have been addressed:
1) Referring to the skin of the loach, was it tested immediately after taking off from the abdomen and the mucus removal? Or, on the contrary,
was it stored in the refrigerator for some period? In case, did you test if this affected the quality of the skin?
2) How did you check the reliability of your setup? How did you fix the samples?
3) No standard deviation is reported for the experiments. Could you please add this information or a comment about it?
4) Figure 2 (flow channel): from the pump, I saw two arrows that exit the system. I was wonder where is the water source for the pump. Maybe, can be the arrow
from the water tank to the pump that needs to turn the tip? Otherwise, I suggest you add a sentence and clarify this point.
5) no information is reported about material properties of the bionic and the smooth surfaces: how/which did you consider them in the FE simulations?
6) Referring to the numerical simulations, did you perform computational fluid dynamics (CFD) or fluid-structure interaction (FSI)?
If the simulations are CFD, I believe this point also affected the numerical results with respect to the experiments. I believe you should add details about this part.
Reviewer 3 Report
The manuscript reports a combined experimental and numerical investigation of water low over a bionic surface which mimics the scaled surface of Paramisgurnus dabryanus loach. A bionic surface on aluminum is fabricated with the help of CNC machining. Water channel flow tests indicated a maximum drag reduction of 23% with respect to the drag on the reference smooth surface. Further, a CFD study was conducted on a smaller flow domain, with five rows of scales in the flow direction. The CFD results also showed a drag reduction which is qualitatively similar to that seen in experiments. The flow streamlines, velocity, pressure, and shear force were extracted from the simulations, with the help of which the experimental observations were discussed.
The study is interesting, and performed relatively well. However, there are numerous concerns regarding the experimental configuration, experimental results, CFD set-up, and the explanation of the experimental observations from CFD results. These are detailed in the 'Comments' below. In additions, some minor comments are also included in the 'Comments'. The current version of the manuscript cannot be accepted, and I suggest a major revision of the manuscript addressing these concerns.
Comments:
- Please check and correct the spelling of Paramisgurnus dabryanus throughout the manuscript.
- Line 15: 'k-e' should be replaced with 'k-ε'.
- Line 34: Ref. 5 should be Ref. 6.
- Line 38: Does adding micro-structure affect its structural characteristics (for example, strength of the material)? Please comment.
- Line 49: Replace 'and etc' with 'etc.'.
- While citing references, the author name is not cited correctly. Please refer the standard format and change this accordingly throughout the manuscript.
- Line 70: Did the maximum drag reduction rate increase by 25.7% or was the maximum drag reduction rate 25.7%? Please check and correct!
- Line 78: What does this mean? Was the ambient flow medium composed of both water and oil? Or, was the ambient flow medium composed of water and the fish scale impregnated with lubricant oil? Please clarify in the manuscript.
- Line 90: Replace 'in the' with 'in'.
- Figure 2: Please specify the cross-sectional area of the flow test section. Secondly, were the dimensions of the samples used for the tests the same as specified in line 100? Please clarify this in the discussion.
- Figure 2: What is the need for the flow line below the line with the samples? Why is the flow direction marked from the pump to the water tank in the bottom most line?
- Figure 3: Figure 3 caption: The caption does not reflect what is shown (schematic diagram of the pressure data acquisition in the test section of the flow channel). Please modify the caption.
- Equations (1), (2): Are these pressure drop calculations from measurements where two identical samples were placed in the test section? For example, does PDsmooth correspond to the difference between Pin and Pout when two smooth samples were placed in the test section? Please clarify this in the manuscript.
- Sec. 3.2: Sec. 3.2: Why is the fabrication of the sample included under 'Results'? Please move this to the previous section where experimental details are mentioned.
- Line 143: How is the shape parameter defined? And, how is this different from that on the loach scales? Please clarify in the manuscript.
- Fig. 5b,c: Fig. 5: Is the surface of the sample, as shown here, not flat? Please clarify this.
- Table 1: 'Flowmeter' should be renamed as 'Flow rate' or 'Volume flow rate'. 'Flow Rate' should be renamed as 'Average flow velocity'. Please specify how the flow velocity was calculated. Please also specify the location along the flow line where the flow rate measurement and flow velocity were obtained.
- What is the typical range of swimming speed of Paramisgurnus dabryanus loach? This could be mentioned in the Introduction.
- The phrases 'pre-smooth/pre-bionic' and 'post-smooth/post-bionic' are not appropriate since they mean the position of something with respect to the samples and not the positions of the samples themselves. These terms have to be re-phrased.
- Figure 6: Fig. 6: The deviation or the range of pressure drop from the repeatability tests could be shown as error bars in this plot.
- Figure 6: It is not meaningful to compare the pressure drop between the curves 1 and 2 since the flow condition immediately upstream of 2 and 4 (between 1 and 2, and between 3 and 4, resp.) are different. Due to this, it only make sense to compare the pressure drop between the curves 1 and 3.
- Figure 6: Why does the smooth surface (1) has smaller pressure drop than the bionic surface (3) at low flow velocities in Fig. 6, whereas Table 1 shows otherwise at these velocities? The same is more pronounced when one compares between 4 and 2. Why is this so?
- Line 169: After 2.6 m/s is not shown in Fig. 6. Hence this statement is not meaningful.
- Line 172: It is not possible to talk about drag reduction, from Fig. 7, without a reference study where both the samples were smooth.
- Figure 7: The colors seem to be interchanged (relative to those shown in Fig. 6). Please check and correct.
- Figure 8 could be a part of the discussion on the artificial sample used in experiments. The choice of the shape of the scales is made clear in Fig. 8. And, this is not clear in the discussion on experiments in the present form.
- Figure 9: Please give the dimensions of the bionic sample used in the simulation.
- Line 196: What is the unit of the time step? Please indicate.
- Equation 3: Please show the details of how this expression is obtained in the response.
- How was the initial/inlet flow velocity set up? Please include this in Sec. 3.4.2.
- Line 206: 'triangular' -- this is different from what is written in Fig. 10. Please check and correct.
- Line 209: 'more intensive meshes' could be replaced with 'the finest meshes'.
- Table 2: How can the flow resistance of S-wall be compared with that of the B-wall? The absence of a smooth sample of the same dimensions as the bionic sample makes this not a meaningful comparison. Secondly, a meaningful comparison is possible only when the bionic sample is replaced with a smooth sample of the same macroscopic dimensions on the same location in an independent simulation run. The modification of velocity profile by the bionic sample make the calculations on the S-wall dependent on the presence of bionic sample.
- Please pecify the units of resistance in Table 2.
- How were the pressure and viscous resistances of the B- and S-walls calculated from the simulations?
- Why was the pressure resistance on the B-wall non-zero? Was this due to the presence of the sample height as a step, which is absent on the S-wall? Please clarify.
- Line 246: Replace 'on the increase' with 'more than that on the smooth sample'.
- Figure 15: Mention the units of the forces expressed here. Secondly, the values of shear force on the sample surface seems to be lower than that of the pressure. Why is this so?
- Figure 16: Why are shear and pressure forces expressed in Pa? Please check and correct. Secondly, please mark the position of the bionic sample in the plots.
- Lines 300-302: How is the pressure resistance on the bionic sample less than that on the smooth sample? This is in contradiction to the data shown in Table 2.
- Lines 308-313: Why does low pressure area mean a better drag reduction? Do the authors mean pressure drag acting on the sample here? And, is the pressure drag dependent on pressure or *differences* in pressure? Lines 311-313 seem to indicate that if one has a higher pressure difference between an upstream location and a downstream location for flow around a body, drag is somehow less -- please explain this in terms of the simple case of flow around a sphere.
- Figure 19: How are the yellow regions identified? Please give the methodology by which these yellow rectangles are marked. Also, please provide a quantitative value for the reduction in the contact area.
- Line 343: pressure and velocity *gradients* are shown in Fig. 20b,c, not pressure and velocity. Please correct the statement in this line accordingly.
- Figure 20: The relevance of these plots is not clear. Please explain.
- Figure 20: In which direction are the gradients calculated -- along the wall or normal to the wall? Please clarify in the manuscript. Secondly, please give the units in the plots. Finally, please replace 'derection' with 'direction' in the schematic picture.
- Figure 21: Between the two scales, the streamlines seem to strongly deflect towards the right (with respect to the flow direction) in 'b' whereas in 'a' such a strong deflection is not seen. Please explain.
- Difference between Fig. 22 and Fig. 23: In the study reported in literature (Fig. 23), the high-speed streaks are seen in the flow passing through overlapping scales, where as in Fig. 22 from the current study the flow along the overlapping region is not shown, all the five lines pass through non-overlapping regions. Please clarify this.
- Figure 23 caption: please check and correct the reference number.
- Lines 375-378: This is due to the spatial resolution of the simulations and not due to the small fish scales. Please indicate this.
- Figure 24a: Why do the scales show undulations at their border and not a smooth border? Secondly, if Y is in the direction normal to the B-wall and the flow is along X direction (as shown in Fig. 9), what does the arrangement of the scales in the XY plane mean? Figure 9 shows the scales are arranged in the XZ plane. Please clarify.
- Figure 24b: Why are there two peaks for inlet, center-line, and outlet cases which have 3, 2, and 3 scales respectively? The 'outlet' symbol color could be made blue.
- Lines 391-394: The presence of low velocity regions does not necessarily mean a reduction in turbulence, since vorticities/eddies present in turbulent flows may exhibit low velocity regions. Hence this statement is not accurate.
- Lines 394-396: The relevance of Refs. 41 and 42 to support this statement is not clear, since these studies deal with lubricating flows between mechanically moving parts.
- Line 398: 'resistance' could be replaced with 'underwater flow resistance'.
- Ref. 11: The author names are not cited correctly here.
Round 2
Reviewer 1 Report
Please take into account my two comments and I will accept your paper.
Figure 14: I think the vertical axis is necessary.
Figure 22b: It's difficult to understand, so please devise an expression.
Reviewer 2 Report
I am satisfied with the answers and how the
authors significantly improved the presentation
and quality of their work. For this reason, I do not
have any other comments and agree with the
publication.
Reviewer 3 Report
The authors have addressed most of the concerns raised in the previous review of this manuscript -- I appreciate the authors' efforts. However, there are still some pending issues which should be addressed in a further 'minor' revision of the manuscript. Of particular concern are the responses to points 36 and 41 in the previous review, which are crucial to the meaningfulness of the results from this study (see points 6 and 8 below).
- The name of the loach is still incorrectly spelled -- it should be Paramisgurnus and not Paramisagurnus -- please change this throughout the manuscript.
- Response to point 6: The authors might have misunderstood this. Citing the references in the *text of the article* (for example, in the Introduction and other sections) should be re-checked. Perhaps, the authors/editors could have a look at it so that the standard format of citing the references in the text of the article could be ensured.
- Response to point 16: Perhaps, the authors missed to address this -- is the surface of the 'artificial bionic sample' also not flat, and how is this flatness measured/compared with that on the real loach surface?
- Response to point 17: Please check the formula for Q in terms of v -- this is incorrect! The factor t (time) should not be present in this formula. Please make appropriate changes.
- Response to point 30: It is not clear what kind of velocity profile was used at the inlet. Was it a uniform 'plug' flow profile or a fully-developed velocity profile, or something else? Please clarify this in the revised manuscript.
- Response to point 36: This is a critical point, since the drag force and its reduction are dependent on the 'differential pressure resistance' as well. Since the difference in this differential pressure resistance between smooth (S) and bionic (B) walls is due to the step height of the B-wall, which is absent in the S-wall, this cannot be attributed to the difference in the surface characteristics (smooth versus bionic). Hence, the arguments on drag reduction deduced from this measurement are questionable. I would suggest to remove the total drag reduction (for example, in Table 4), since this is an incorrect comparison. Or, at the least, the authors should indicate that this total drag reduction, which includes the pressure drag reduction as well, does not highlight the effect of surface characteristics alone (due to the difference in step height on the B- and S-walls). This is evident as the pressure drag on the B-wall increases by almost three orders of magnitude as the Re is increased, whereas that on the S-wall remains unchanged at zero.
- Response to point 39: Please replace 'pressure' by 'pressure force' in the figure. Similarly, in Fig. 19, if the plot is pressure then the unit should be Pa, or if the unit is N then it should be 'pressure force'. Please correct.
- Response to point 41: Due to the presence of step height on the B-wall and its absence on the S-wall, it is unclear if the low pressure regions seen on the B-wall are due to the step height of the B-wall or due to the surface characteristics. It is difficult to differentiate how much these two causes (step height and surface characteristics) contribute to the observed reduction in viscous drag. Some discussion on this could be helpful for the readers.
